# Techno-Economic Optimization of a Solar–Wind Hybrid System to Power a Large-Scale Reverse Osmosis Desalination Plant

**A. M. Soliman** [1,2,*]**, Abdullah G. Alharbi** [3] **and Mohamed A. Sharaf Eldean** [2]

1 Department of Mechanical Engineering, College of Engineering, Jouf University, Sakaka 72388, Saudi Arabia
2 Department of Mechanical Engineering, Faculty of Engineering, Suez University, Suez 41522, Egypt; swazy20@hotmail.com
3 Department of Electrical Engineering, College of Engineering, Jouf University, Sakaka 72388, Saudi Arabia; agalharbi@ju.edu.sa
* Correspondence: amsoliman@ju.edu.sa

**Abstract:** Solar-wind hybrid systems have grown to become a pivotal option for powering membrane desalination processes, especially because they have zero harmful emissions. In this work, solar photovoltaic (PV) and horizontal wind turbine (HWT) systems were used to drive a reverse osmosis (RO) desalination process to produce large-capacity fresh water. Moreover, an investigation into a hybrid PV–HWT system combined with RO was also conducted. The proposed systems are compared technically and economically with the solar organic Rankine cycle (SORC) for RO. Technical and analytical optimization methods were performed to minimize the unit product cost (USD/m$^3$). The results revealed that photovoltaic-powered RO is recommended over wind energy operations. However, for large capacities, both thermal and wind farms dominate.

**Keywords:** photovoltaic system; horizontal wind turbines; solar organic Rankine cycle; reverse osmosis

## 1. Introduction

The Middle East and North Africa (MENA) have the world's lowest per capita availability of water resources [1]. Fortunately, most MENA countries have a renewable energy potential that encourages the application of solar and wind to drive desalination units. Renewable energy to drive the desalination units is a vital solution to water scarcity in remote areas that lack conventional energy sources like heat and electricity grid [2]. Reverse osmosis (RO) is a modern process to desalinate water for a wide range of applications [3].

For this technique, mechanical energy in the form of a high-pressure pump, not thermal energy, is required. In recent decades RO has gone through a remarkable development. The number of RO plants has increased and the capacity of the units has reached 3,000,000 m$^3$/day [4]. The reliability and the availability of combining renewable energy resources such as solar and wind power are reasons for using RO instead of thermal distillation processes. Photovoltaic (PV) panels are the common form of solar energy generation in RO units. Essam Mohamed et al. [5] investigated technically and economically a photovoltaic system to drive RO desalination. The productivity of that system was 0.35 m$^3$/day with a specific power consumption of about (4.6 kWh/m$^3$). The cost of fresh water was about (EUR 15–20/m$^3$), which is high because of the need for a large number of batteries to stabilize the pressure and flow rate for the membranes [5]. An economic feasibility study of PV power on RO within low specific power consumption was established by Helal [6]. It was conducted using three alternative configurations of an autonomous PV–RO unit in the United Arab Emirates (UAE). In that study, a diesel generator was used during periods of low sunlight. The productivity of PV-RO was not more than 20 m$^3$/day (10 h). In Helal's study, the environmental impacts of diesel generator emissions were ignored. The technical characteristics of PV-RO desalination systems and an economic comparison were presented by Manolakos [7]. The total peak power of the PV system was 846 W and consisted of

18 Arco-Solar mono-crystalline PV panels. The productivity of the system was 0.1 m$^3$/h and the specific energy consumption has been found to be in the range of 3.8–6 kWh/m$^3$. It is found that the cost of 1 m$^3$ of fresh water was about EUR 7.77. Ahmed [8] found that the cost of the productivity for a PV-powered small-scale RO water desalination system is USD 3.73/m$^3$. In Morocco, Tzen [9] studied an autonomous PV–RO system for the potable and other water needs of a rural community and found that the specific power consumption (SPC) was 15 kWh/m$^3$).

Wind energy is also used in this kind of operation, and some of the literature is provided in this section. Liu et al. [10] studied a RO system driven by wind energy for wastewater treatment without an economic study. An experimental RO plant connected directly to a wind system without energy storage was presented by Pestana [11]. The system was designed based on 21 kW of power, to produce 3.6 m$^3$/h. Dehmas et al. [12] found that wind energy can successfully power a desalination plant. Their studies included the economic analysis of 5 Bonus 2 MW wind turbines. Garcia-Rodriguez et al. [13] presented the influence of the climate, nominal power of the wind turbine, salt concentration, design arrangement, operating conditions, and plant capacity on the cost of fresh water. Romero-Ternero et al. [14] presented exergo-economic analysis for wind-powered seawater RO system the unit cost of freshwater was EUR 0.76/m$^3$. Wind RO plants have a wide range capacity of 12–2500 m$^3$/day with nominal power of 4–200 kW [15]. Dimitriou et al. [16], and Ruiz-García and Nuezb [17], investigated the performance of RO under variable operating conditions, and another study by Ruiz-García and Nuezb [18] investigated the long-term intermittent performance of a brackish water RO desalination plant.

From the literature review, the hybrid wind–PV system for driving RO has shown advantages: a wide range of capacities, minimal energy loss, easy maintenance and reliability. However, such systems were not investigated from a problem optimization or techno-economic perspective. Optimization criteria are very important because they reduce the total power consumption on the RO high-pressure pump (HPP), thereby, reducing the area of the solar field and the total cost. It is obvious from the literature that the possibility of producing a large water capacity (>3000 m$^3$/day) is far away because of the cost limitations of PV and wind technologies. Furthermore, electric storage for PV and wind power is still unaffordable. In this study, under the steady regime, the design and optimization of the various capacities of a PV-horizontal wind turbine (HWT)–RO system are investigated to come to a clear decision about the feasibility of using PV and HWT regardless of location or weather condition. Moreover, the comparison between the proposed systems and the solar thermal power cycle of a RO operation is performed. The optimization of techno-economic issues is the focus of this work to decrease power consumption and system costs. The work proceeds as follows:

- Process configurations are presented, design limits are investigated, and the mathematical model of the proposed systems is presented.
- The optimization method is performed to study the effect of cost minimization.
- A comparison of four cases (solar direct–three configurations vs. solar indirect–one configuration) is performed.

## 2. The Process Configurations

The combination of solar power with RO was achieved by one of two methods. The first is electrical (PV and HWT), and the second is via the thermal solar Rankine cycle. The proposed configuration related to the electrical method was modeled using SDS software and shown in Figure 1 [19]. SDS is a developed software library as a part of the REDS program library developed by Sharaf et al. [16]. The model configuration contained a PV, HWT, inverter unit, battery bank (storage unit) and a control room for power switching between PV and HWT to operate the RO system.

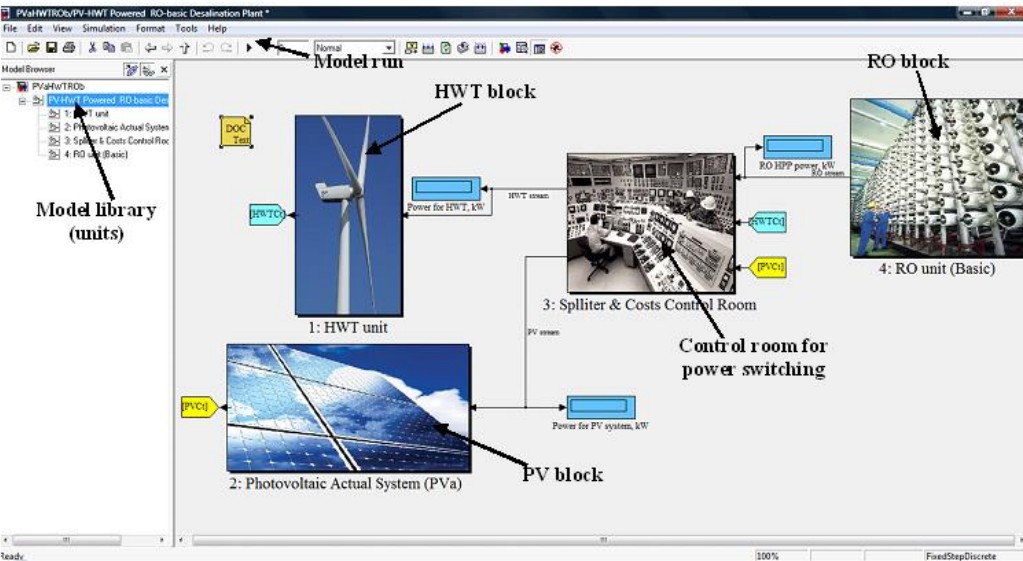

**Figure 1.** The proposed system model browser of the PV–HWT–RO using the SDS program. The system contains: 1, HWT; PV system; 3, control room unit; 4, RO desalination plant block.

The RO unit was operated by HWT, PV or both (PV–HWT). Concentrated solar power (CSP) is shown in Figure 2. This configuration is presented based on thermal power to operate the organic Rankine cycle (ORC) to generate power to operate the RO system. This configuration contained a parabolic trough collector (PTC) field for the thermal power, a boiler heat-exchanger unit (BHX), pumps, ORC turbine, recuperator for regeneration and energy recovery, a condenser unit for heat rejection and the pre-heating of salt water stream and the RO system. Therminol-VP1 [20] was used as heat transfer oil (HTO) in the fourth configuration. Toluene was used as the working substance through the ORC [2,20,21].

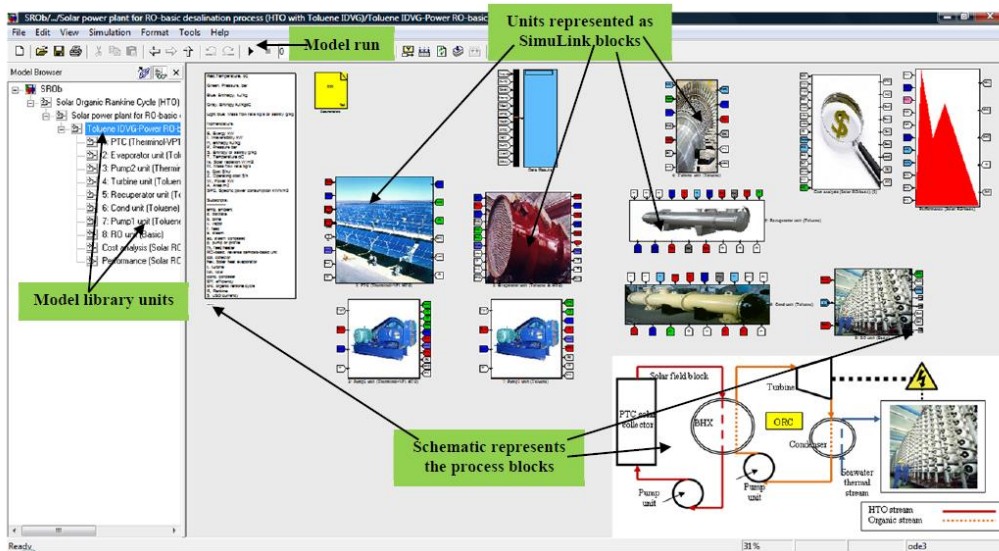

**Figure 2.** Solar ORC model under SimuLink software browser by the use of SDS program: PTC, field, BHX unit, pumps, turbine unit, heat exchanger recuperator and RO plant block.

## 3. Simulation Methodology

The calculated parameters were area, dimension, mass flow rate, and temperature. For the desalination, the fresh water capacity was specified, after which the electrical load was calculated for either PV or the HWT design specifications. The calculated parameters

and design limits of the proposed units are illustrated in Table 1. The mathematical model representing the process configurations and validity is listed in the Appendix A.

**Table 1.** The specified parameters based on the design technique of modeling concept.

| RO Model | |
| --- | --- |
| **Specified** | **Calculated** |
| Fresh water productivity, m$^3$/day | Feed and brine mass flow rates, kg/s |
| Seawater temperature, °C | Pressure on the HPP, bar |
| Seawater salinity, ppm | Average pressure, bar |
| HPP efficiency, % | Product and brine salinities, ppm |
| Booster pump efficiency, % | Salt rejection percentage, % |
| Membrane fouling factor, % | HPP power, kW |
| Number of elements/number of pressure vessels | SPC, kWh/m$^3$ |
| Element area, m$^2$ | Membrane area, m$^2$ |
| Recovery ratio, % | |
| Pressure exchanger (PEX) efficiency, % | |
| **HWT Model** | |
| **Specified** | **Calculated** |
| Power/set, kW | Starting and average wind speeds, m/s, and air mass flow, kg/s |
| System total power, kW | Rotor diameter, m |
| Ambient temperature, °C | Hub height, m |
| Ambient pressure, bar | Rotor speed, rpm |
| | Swept area, m$^2$ |
| | Axial force, kN and Torque, kN |
| | Power coefficient, % |
| | Number of wind turbines |
| | Spacing between turbines in winddirection, m |
| | Spacing cross turbines in wind direction, m |
| | Total farm area, km$^2$ |
| **PV Module** | |
| **Specified** | **Calculated** |
| Solar radiation, W/m$^2$ | The open circuit voltage, V and the short circuit current, A |
| Power/panel, W | The maximum voltage and current |
| Total system power, kW | The cell and module efficiencies, % |
| | The number of cells and modules of the system |
| | The module and system weights, kg, and areas, m$^2$ |
| | The battery bank capacity, A |

**Table 1.** *Cont.*

| S-ORC Model | |
|---|---|
| **Specified** | **Calculated** |
| Solar radiation, W/m$^2$ | Solar Fields, condenser, recuperator areas, m$^2$ |
| Parabolic trough collector (PTC) high temperature, °C | Solar field dimensions and design |
| Boiler heat exchanger (BHX) effectiveness, % | Mass flow rates, kg/s |
| ORC turbine efficiency, % | ORC mass flow rates, kg/s |
| Recuperator effectiveness, % | Solar field mass flow rate, loop design, No. of collectors |
| Condensation temperature, °C | Heat rejected power, kW |
| Condenser effectiveness, % | PTC pump power, kW |
| | All thermo physical properties of all streams |

## 4. Results and Comments

### 4.1. Non-Optimization Comparison Results

There are many ways to judge process performance results such as by field area, fresh water production rate, salinity range or specific costs. The main performance indicator studied in this work was the unit product cost (UPC) in USD/m$^3$. This indicator is very important, giving an early evaluation before the optimization procedures. In this section, the process results are obtained at an RO production rate of 3500 m$^3$/day as a pre-stage before optimization. The costs of the system units are illustrated in the Appendix A. The user has to specify total productivity and then all the required parameters will be calculated relatively and iteratively.

Table 2 shows the data results for the HWT-RO system for 3500 m$^3$/day. The results revealed that to generate 1131 kWe, two 600-kWe HWTs were used. HWT-600 kW was used because the expected wind speed in the Suez Gulf region is suitable. The average wind speed is 11–12 m/s. Technical analyses showed that the hub height was about 41 m with a 43 m rotor diameter. The wind farm occupies about 0.063 km$^2$ of land; the total annual cost is about 6.31 × 10$^5$ USD/year; and the unit product cost (UPC) is about 0.5541 USD/m$^3$.

Table 3 presents the results of the PV–RO system that produced the same amount of fresh water (3500 m$^3$/day). The expected power load was 1131 kWe for the RO plant, and a 220 W module is chosen for the PV site. In addition, 350 W/m$^2$ of solar radiation was chosen as a known input parameter to calculate the PV site area based on the worst winter conditions [22]. Excess power in summer could be stored in other facilities such as the control room cabinet or lighting issues in the plant. Compared to the HWT–RO, the PV–RO had a lower UPC (USD 0.53 vs. 0.55/m$^3$) and a smaller solar field site (about 0.02 vs. 0.06 km$^2$). Based on current results, the PV–RO system is considered to have a lower UPC than the HWT–RO system.

Table 4 shows the results of the hybrid system in which the power demanded by the RO plant is equally divided between the HWT farm and the PV site. The results revealed that the hybrid took up a larger area (HWT farm and PV site area) and the UPC increased to USD 0.5744/m$^3$ due to the HWT number increase. It is obvious from the related tables that the UPC could be optimized to a minimum value with respect to the site area by selecting the right PV module power or the number of wind turbines. Moreover, the PV–RO system was considered attractive against the remaining systems according to the minimum area required and UPC in USD/m$^3$. However, the 280 W module may consume a larger area with a higher cost per site. Therefore, optimization should be implemented to identify the greatest effect of each system on the UPC in USD/m$^3$.

**Table 2.** Preliminary data results for HWT–RO system for 3500 m$^3$/day.

| Environmental Conditions | |
|---|---|
| Ambient temperature, °C | 15 (winter) |
| Solar radiation, W/m$^2$ | 350–400 (winter) [22] |
| Air pressure, bar | 1.01 |
| Seawater temperature, °C | 20 |
| **RO Plant Results** | |
| Specific power consumption, kWh/m$^3$ | 7.68 |
| Power, kW | 1131 |
| Feed mass flow rate, m$^3$/h | 485.9 |
| Production flow rate, m$^3$/h | 145.83 |
| Brine flow rate, m$^3$/h | 340.1 |
| Brine salinity, ppm | 64180 |
| Fresh water salinity, ppm | 250 |
| Salt rejection value | 0.9944 |
| RO high pressure, kPa | 6850 |
| **HWT Farm Results** | |
| Total power for RO plant, kW | 1131 |
| Turbine power/wind power, kW | 600/4762 |
| Starting wind speed, m/s | 4–11.95 |
| Rated wind speed, m/s | 17.62 |
| Hub height, m | 41.6 |
| Rotor diameter, m | 43.32 |
| Swept area, m$^2$ | 1474.08 |
| Air mass flow rate, kg/s | $3.173 \times 10^6$ |
| Torque (Nm)/rpm | $1.542 \times 10^5$/37.15 |
| No. of wind turbines | 2 |
| Farm area, km$^2$ | 0.06368 |
| **Cost Results** | |
| Plant life time/interest rate | 25/5% |
| DCC of wind turbines, USD | $1.174 \times 10^6$ |
| DCC of RO plant, USD | $3.5 \times 10^6$ |
| Annual total costs, USD/year | $6.31 \times 10^5$ |
| Unit product costs, USD/m$^3$ | **0.5541** |

**Table 3.** Preliminary data results for PV–RO system for 3500 m$^3$/day.

| Environmental Conditions | |
|---|---|
| The environmental conditions | Presented in Table 2 |
| **RO Plant Results** | |
| The RO results | Presented in Table 2 |
| **PV Site Results** | |
| Open circuit voltage/short circuit current, V/A | 58.6/8.49 |
| Maximum voltage/maximum current, V/A | 47.4/4.641 |
| Module efficiency/cell efficiency, %/% | 15.45/17.5 |
| No. of cells per module/No. of total modules | 96/5135 |
| Module dimensions/width, m$^3$/mm | 0.095/45 |
| Net weight, kg | 21.5 |
| Cell area/module area, cm$^2$/m$^2$ | 423.8/4.06 |
| Total system area, km$^2$ | 0.02089 |
| Battery storage, Wh | $3.766 \times 10^6$ |
| Battery capacities, Ah | $7.945 \times 10^4$ |
| No. of batteries—12 volt system | 4 |

**Table 3.** *Cont.*

| Cost Results | |
|---|---|
| Plant life time/interest rate | 25/5% |
| DCC of PV, USD | $7.912 \times 10^5$ |
| DCC of RO plant, USD | $3.5 \times 10^6$ |
| Annual total costs, USD/year | $6.099 \times 10^5$ |
| Unit product costs, USD/m$^3$ | **0.5305** |

**Table 4.** Preliminary data results for HWT–PV-RO system for 3500 m$^3$/day.

| Environmental Conditions | |
|---|---|
| The environmental conditions | Presented in Table 2 |
| **RO Plant Results** | |
| The RO results | Presented in Table 2 |
| **HWT Farm Results** | |
| Total power for RO plant, kW | 1131 |
| Turbine power/wind power, kW | 100/596.2 |
| Starting wind speed, m/s | 5.6 |
| Rated wind speed, m/s | 14.77 |
| Hub height, m | 20.03 |
| Rotor diameter, m | 19.64 |
| Swept area, m$^2$ | 303.1 |
| Air mass flow rate, kg/s | 5467 |
| Torque (Nm)/rpm | $1.288 \times 10^4$/74.14 |
| No. of wind turbines | 6 |
| Farm area, km$^2$ | 0.03924 |
| **PV Site Results** | |
| Open circuit voltage/short circuit current, V/A | 58.6/8.49 |
| Maximum voltage/maximum current, V/A | 47.4/4.641 |
| Module efficiency/cell efficiency, %/% | 15.45/17.5 |
| No. of cells per module/No. of total modules | 96/5135 |
| Module dimensions/width, m$^3$/mm | 0.095/45 |
| Net weight, kg | 21.5 |
| Cell area/module area, cm$^2$/m$^2$ | 423.8/4.06 |
| Total system area, km$^2$ | 0.01045 |
| Battery storage, Wh | $1.88 \times 10^6$ |
| Battery capacities, Ah | $3.97 \times 10^4$ |
| No. of batteries (12 volt system) | 4 |
| **Cost Results** | |
| Plant life time/interest rate | 25/5% |
| DCC of HWT, USD | $1.1 \times 10^6$ |
| DCC of PV, USD | $3.958 \times 10^5$ |
| DCC of RO plant, USD | $3.5 \times 10^6$ |
| Annual total costs, USD/year | $6.604 \times 10^5$ |
| Unit product costs, USD/m$^3$ | **0.5744** |

*4.2. Comparison Results Based on Optimization*

It was becoming very hard to recognize the main cause for the UPC. Previous results indicated that the PV–RO system had the lowest UPC; however, such a result needs more investigation based on many factors such as the number of wind turbines, modules of watts of PV, and RO operating conditions. For that purpose, technical and analytical optimization techniques were implemented. The technical method was implemented for the RO part, but the analytical method was implemented for the RO, HWT, and PV systems.

### 4.2.1. Technical Optimization Results

In this section, the solution to minimize the objective function (UPC, USD/m$^3$) was implemented for the RO part. Optimizing it influenced the HWT and PV. To minimize the UPC, it was necessary to reduce the HPP power load and maintain the same fresh water production rate. Increasing the number of stages decreased the HPP power load. For the same case study (3500 m$^3$/day), the number of stages was increased to nine. Table 5 shows that the increased number of stages decreased the power from 1131 to 917.47 kW (18–20%). Then the calculated SPC became 6.3 vs. 7.7 kWh/m$^3$ in the basic case. Based on a previous study [3,21], increasing the number of stages decreased the HPP power load and the PEX device dominated operations in RO plants, reducing power consumption by 60–65% [22]. Table 5 shows the data results of the RO plant for 3500 m$^3$/day based on the different technical devices. Therefore, the PEX device is recommended in this study.

**Table 5.** Results comparison for different energy recovery devices.

| Parameter: | Power, kW | SPC, kWh/m$^3$ | RO ΔP, bar | Power Reduction, % |
|---|---|---|---|---|
| Basic | 1131 | 7.7 | 68.66 | – |
| Stages = 9 | 917.47 | 6.3 | 35.8 | 18–20% |
| PEX | 380–394 | 2.704 | 68.74 | 60–65% |

### 4.2.2. The Analytical Optimization Results

Based on the technical results in the previous section, the RO-PEX technique is recommended for the analytical method. Figure 3 shows the effect of the RO-PEX inlet feed/splitter ratio and the total system productivity on the UPC. The figure addressed the case of HWT-RO-PEX without the operation of the PV solar field. Decreasing the inlet feed splitter ratio to 10% increased the demanded power by the HPP by more than three times (390 to 1532 kW). The UPC was directly proportional to this effect, increasing to 0.683 USD/m$^3$ at 3500 m$^3$/day. Furthermore, the effect on total productivity was also notable with respect to the feed/splitter ratio. The number of HWTs in the wind farm was a very important parameter that affected the UPC (Figure 3b). Increasing the number of wind turbines (low power per unit) increased the UPC. Therefore, fewer than two units are recommended for this operation.

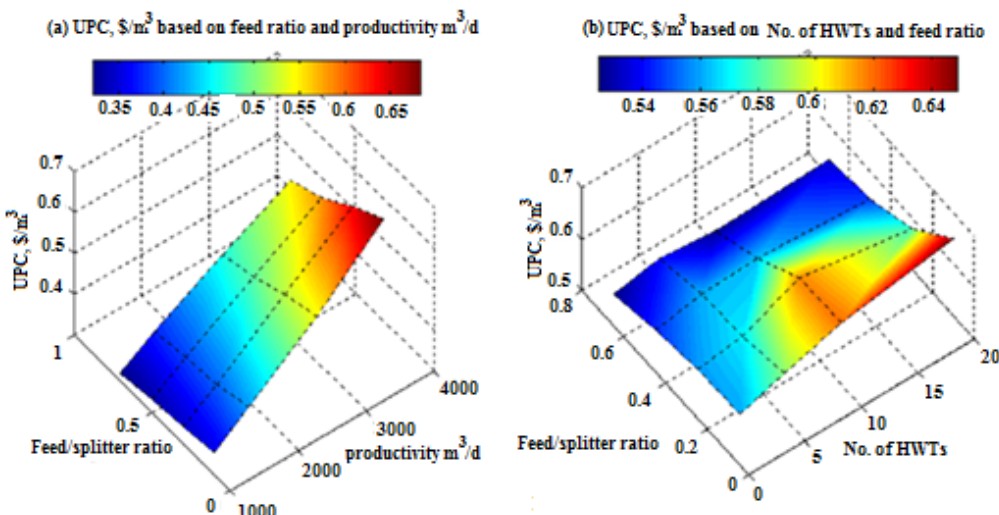

**Figure 3.** (**a**) Effect of feed splitter ratio and RO-PEX productivity on the UPC, USD/m$^3$; (**b**) effect of feed/splitter ratio and no. of HWTs on the UPC, USD/m$^3$.

Decreasing the quantity of total fresh water from 3500 to 1000 m$^3$/day would also increase the UPC for the same operating conditions. At 3500 m$^3$/day and a feed/splitter ratio of 70%, the UPC is about 0.548 USD/m$^3$. Figure 4a,b represents the effect of the

inlet feed splitter, PV module power, and total productivity on the UPC. Figure 4a,b demonstrates the effect on the UPC without the operation of the HWT farm. In Figure 4a, results are obtained at total productivity of 3500 m$^3$/day. The figure shows that it was important to increase the feed/splitter ratio related to the RO-PEX part because it decreased the HPP power demand and hence lowered the UPC. At the same time, selecting a module with a higher rate of power increased the UPC. The results revealed that the PV 35 W module was the best choice with a feed/splitting ratio of more than 70%. However, the PV 280 W module increased the UPC. For a ratio of 70% and PV 35 W, it was about 0.517 USD/m$^3$.

Figure 4b shows the effect of various system production rates from 1000 to 3500 m$^3$/day on the UPC. The optimized point is obvious at 0.31 against 0.557 USD/m$^3$. Decreasing total productivity while increasing the inlet feed/splitter ratio led to a decrease in the UPC. For the operation of the PV field, the 35 W module is effective because of the individual unit price. Moreover, increasing rather than decreasing the inlet feed/splitter ratio is recommended for the following reasons:

- It decreases the electrical load on the HPP unit;
- The operation proceeds without any need for more pressure vessels or membranes;
- The gain power is fully loaded on the PEX unit without any excess power from the brine loss.

Suppose that the investor or the designer had to choose between the PV solar field or the HWT farm or both with respect to the control room load (CRL) distribution. From the results, the operation of the PV is recommended over the HWT operation based on the UPC. For both operations, increasing the load percentage meant increasing the dependent load on the HWT farm. Figure 5 shows the variations in the electric load distribution between the PV and the HWT. It is evident that the UPC was minimized by a load percentage of less than 5%; i.e., fully operated by the PV solar field. Decreasing the load on the HWT farm lowered the UPC. The operation with less than a 5% load on the HWT farm was remarkable and cost about USD 0.518/m$^3$. Criteria to consider when selecting the type of operation are

- The PV module power and number of HWTs,
- The total productivity with respect to the inlet feed/splitter ratio, and
- The site operating conditions, which decide the number of HWTs in the farm based on the power of each turbine.

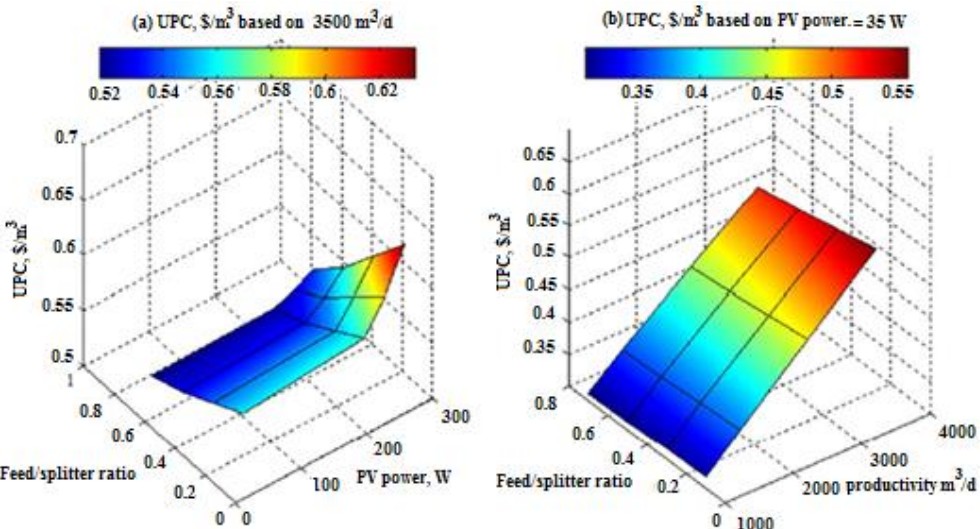

**Figure 4.** (**a**) UPC, USD/m$^3$ vs. PV power, W and RO–PEX feed splitter ratio, (**b**) UPC, USD/m$^3$ vs. productivity, m$^3$/day and RO–PEX feed splitter ratio (PV solar field without HWT farm).

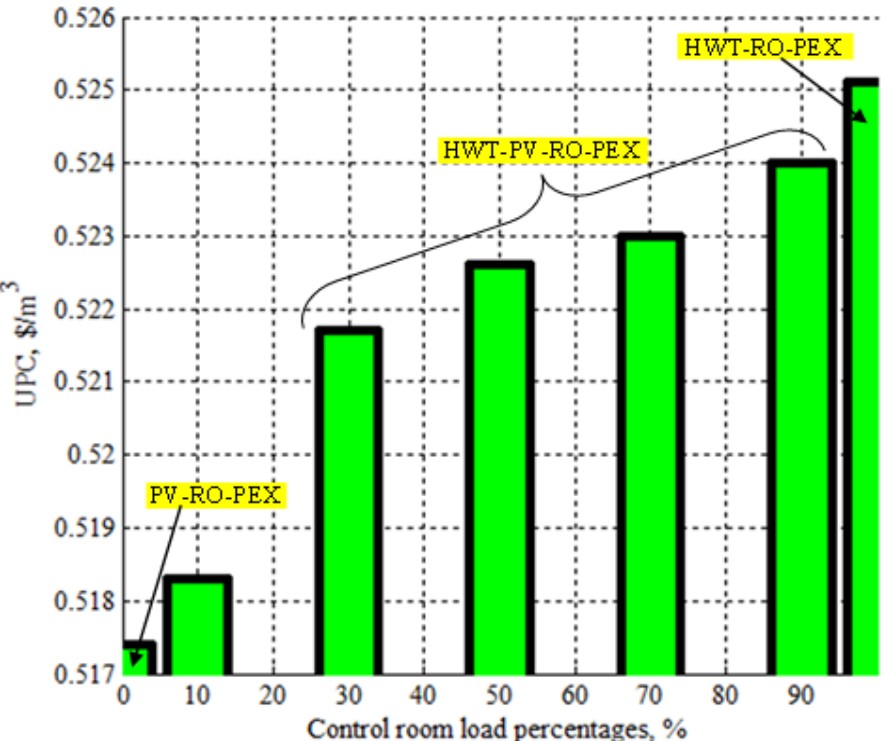

**Figure 5.** The effect of control room load distribution and the UPC, USD/m³ (PV = 35 We module and HWT No. = 1 turbine).

According to Equations (A1) and (A14) in the Appendix A, increasing the power load depends on system productivity, and the feed mass flow rate and the HPP pressure depend on the design parameters of the RO modules. Therefore, the power is a multidimensional functional of parameters:

$$Power = f \begin{cases} \Delta P = f(A_e, n_e, N_V, k_w, FF, TCF) \\ M \\ \phantom{M} f=f\{ \begin{array}{c} M_d \\ RR \end{array} \end{cases} \tag{1}$$

where $A_e$, $n_e$, $N_V$ are the design parameters for the RO, and their effect is notable on the RO pressure.

In this optimization, the constraints of the algorithm are *FF = 0.85; TCF = f(T_f)*; $A_e$ = 35.5 m²; $n_e$ = 8 elements, and the limits of the system are changed around the $M_d$, $N_V$, and the $\Delta P$. The results revealed that more productivity meant more power at the same recovery ratio *(RR)* of 0.3. It was obvious that by increasing system productivity up to 200,000 m³/day the $N_V$ would directly affect the UPC at the same pressure category (e.g., 40 bar). The UPC increased from 0.47 to 7.33 USD/m³ by the use of 20,000 $N_V$. Increasing the $N_V$ decreased the UPC by increasing the $\Delta P$, thereby decreasing the power and the load on the PV site. For moderate production (10,000 m³/day), about 1450 pressure vessels at 40 bar were proposed; however, 750 pressure vessels to produce 100,000 m³/day at 80 bar were used. It is clear now that the choice should have been made according to the lower cost for each pressure category. Therefore, 40 bar should be used, but the massive number of pressure vessels is considered a hindrance to maintenance and system control. For less control and fewer maintenance issues 80 bar is preferred; moreover, maximum production could reach 200,000 m³/day with 1500 pressure vessels compared with 20,000 pressure vessels for 40 bar. According to the PV site, the operation of 200,000 m³/day is considered a massive challenge because of the relation of the site area to the PV site, which would have to generate about 24 MWe. Therefore, because of cost, the 40-bar category was favored in

this study. Figure 6 shows the data results of the effect of pressure and productivity on the system pressure vessel numbers and UPC.

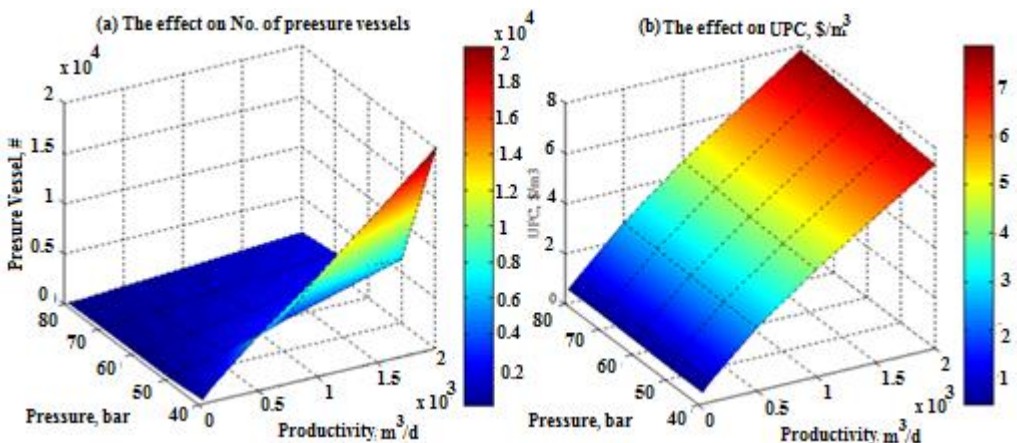

**Figure 6.** Effect of HPP pressure and system productivity on (**a**) No. of pressure vessels, and (**b**) UPC in USD/m$^3$.

### 4.3. Data Results Compared with Solar-ORC

The PV-RO is compared with the S-ORC-RO system under the same operating conditions based on the 3500 m$^3$/day capacity. For HPP$_{RO}$ = 40 bar category, the solar thermal cycle (PTC) consumed a larger area (4475 vs. 3243 m$^2$) and a larger UPC (1.365 vs. 0.4705 USD/m$^3$). Increasing the number of pressure vessels increased the indirect costs for both systems. Moreover, the power load on the thermal units was synchronized with more thermal units, mass flow rates and pumps comparing with direct contact in the case of PV-RO. Even though the HPP$_{RO}$ = 80 bar category harvested more area, the UPC was lower for thermal system; however, it increased a little in the electric one. The HPP$_{RO}$ = 80 bar category required more power, i.e., more area was needed. Figure 7 shows the data comparison between both systems according to the area and UPC at 3500 m$^3$/day. Increasing the pressure on the RO reduced the number of pressure vessels (26 at 80 bar vs. 450 at 40 bar). Therefore, the power increase to harvest more solar area caused an increase in the UPC. Figure 8 shows variations in the solar field area for both sites (thermal and electrical) according to the variations of HPP power. Increasing the power demand by the RO pump required more area to cover the load. The competition of PV with solar thermal for power generation is still far away from implementation because of limitations to the PV operation. Table 6 illustrates the comparison between the PV and the CSP thermal power.

It is clear that the UPC of the PV–RO system was lower than that of the CSP–RO; however, it may increase during operation for high rates of power (over than 10 MWe). because of the high price of the PV panels. Reducing the capital costs of the PV panels may reduce the UPC making it competitive with thermal systems. One possible way to reduce the UPC is to increase the construction rate by building many PV–RO systems with higher rates of productivity for larger remote areas. It may be noted that a PV-powered RO system is suitable for remote regions. Generally, CSP is favorable for larger categories of power generation up to 600 MW.

Figure 9 shows the UPC comparison between Toh et al. [23] and the present work with different feed pressures. Increasing the pressure on the RO caused an increase in the UPC, and the results showed compatibility and convergence.

**Table 6.** CSP vs. PV solar power generation.

| Parameter | CSP (Thermal) | PV (Electric) |
|---|---|---|
| Resource quality | 2400 kWh/m$^2$/year | 2445 kWh/m$^2$/year |
| Power type | Thermal (indirect) | Electrical (direct) |
| Desalination system to combine with | All types (MSF, MED, MED–TVC, MED-MVC, RO, ED | RO, MED-MVC |
| Levelised cost of energy USD/MWh | 60–350 (USD 214 in 2030) | 100–450 (USD 303 in 2030) |
| Construction period/life time | 2/30 years | 1/30 years |
| Capacity factor | 23–50% | 20% |
| Power production | 600 MW | 10 MW |
| Heat engines | Stirling, Rankine, gas turbines, steam turbines | N/A |

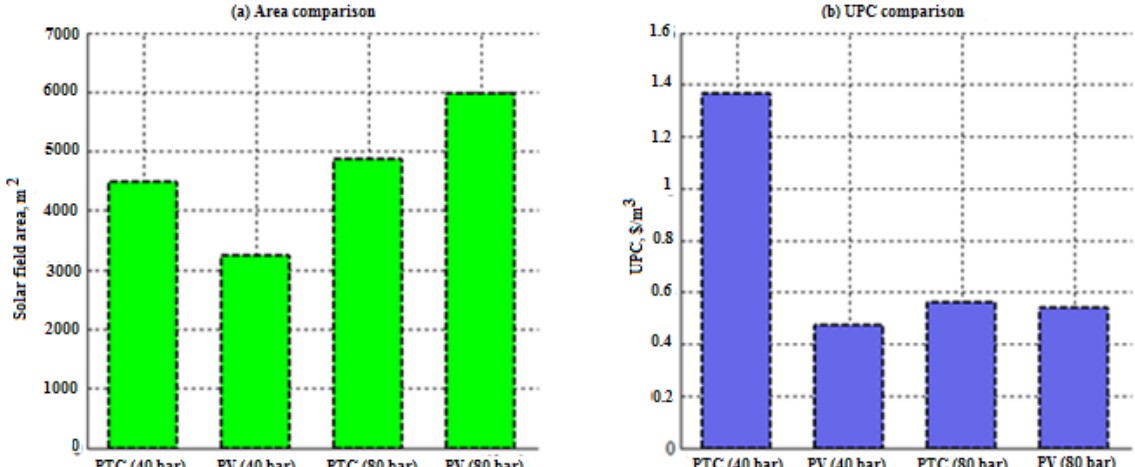

**Figure 7.** Data comparison between PV-RO and S-ORC-RO: (**a**) area comparison, (**b**) UPC comparison.

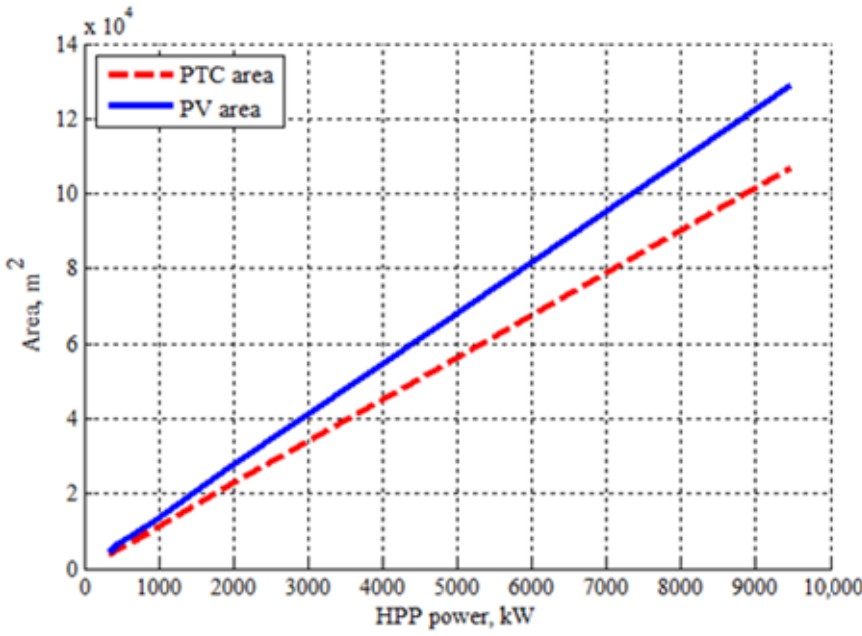

**Figure 8.** The effect of HPP power, kWe on the solar filed area for both techniques thermal and electrical.

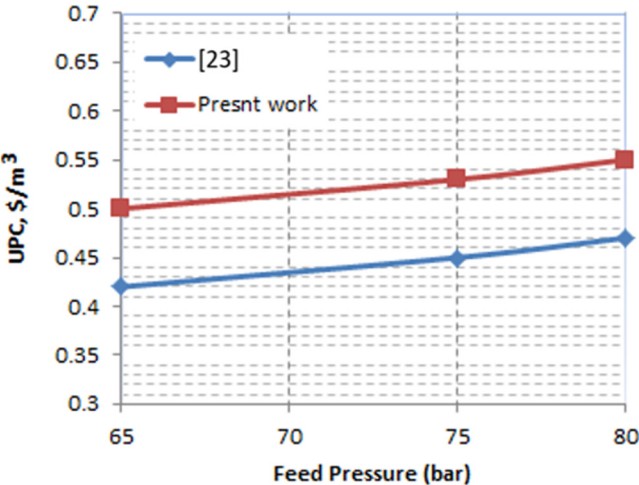

**Figure 9.** Comparison of UPC between Toh et al. [23] and the present work with different feed pressures.

## 5. Conclusions

This work was concerned with powering a reverse osmosis (RO) desalination system by using renewable energy sources such as solar photovoltaic and wind energy. Three configurations for powering the RO were compared analytically by the use of SDS-REDS software. The first was HWT-RO; the second was PV-RO, while the third was the hybrid system HWT-PV-RO. To select the most cost-efficient one to be compared with the solar thermal Rankine cycle (S-ORC-RO), a technical and analytical optimization had to be implemented. The technical optimization recommended the use of the PEX technique, which reduced power consumption by 65%.

For analytical optimization, the following items are recommended:

- Reducing the inlet feed/splitter ratio to 10% to more than triple the demanded power by the HPP (390 to 1532 kW).
- Using fewer than two HWTs for this kind of an operation because increasing the number of wind turbines (low power per unit) increases the UPC.
- Selecting a PV module that minimizes UPC. The results revealed that the PV 35 W module was the best choice with a feed/splitter ratio of more than 70%, whereas the PV 280 W module increased the UPC. The PV 35 W UPC was about 0.517 USD/m$^3$.
- Maintaining less than a 5% load on the HWT farm to minimize UPC. For both operations, increasing the load percentage meant increasing the dependent load on the farm, and this work confirmed that was UPC is minimized when the load percentage was less than 5%; i.e., fully operational by the PV solar field. This operation gave a remarkable UPC of about 0.518 USD/m$^3$.
- Reducing the capital costs of PV panels to reduce the UPC so that it can compete with thermal systems. In a comparison with S-ORC-RO, the UPC of the PV-RO system was lower than that of the CSP-RO system; however, it may increase if the operation generates power over 10 MWe.

Generally, the CSP was favorable for larger categories of power generation up to 600 MW. The PV-RO and HWT-RO were proven promising for producing sustainable fresh water. PV-RO can compete against the HWT-RO or the S-ORC-RO for lower rates of fresh water production. HWT-RO and S-ORC-RO consume larger areas; however, they could be used for higher rates of power and fresh water production.

## 6. Recommendations

A wide range of varied operating conditions should be examined, and system performance should be studied using more design parameter variation.

**Author Contributions:** Methodology: A.M.S. and M.A.S.E.; formal analysis: A.M.S. and M.A.S.E.; software, M.A.S.E.; validation, A.M.S., A.G.A. and M.A.S.E.; formal analysis, A.M.S. and M.A.S.E.; investigation, A.M.S., A.G.A. and M.A.S.E.; resources, A.G.A.; administration, A.M.S.; funding acquisition, A.G.A.; writing—original draft preparation: A.M.S. and M.A.S.E.; review and editing: A.M.S. and A.G.A.; supervision: A.M.S.; project administration: A.G.A. All authors have read and agreed to the published version of the manuscript.

**Funding:** This work is supported by Ministry of Education in Saudi Arabia for funding this work through the project number 375213500.

**Institutional Review Board Statement:** Not applicable.

**Informed Consent Statement:** Not applicable.

**Data Availability Statement:** Not applicable.

**Acknowledgments:** The authors extend their appreciation to the Deputyship for Research & Innovation, Ministry of Education in Saudi Arabia for funding this work through the project number 375213500.

**Conflicts of Interest:** Authors declare no conflict of interest.

## Nomenclature

| | |
|---|---|
| $A$ | Area: $m^2$ |
| $Ac$ | Cell area, $m^2$ |
| $Ae$ | Element area, $m^2$ |
| $AH$ | Battery capacity, Ah |
| $Am$ | Module area, $m^2$ |
| $Ar$ | Rotor swept area $m^2$ |
| $At$ | Total area, $m^2$ |
| $BS$ | Battery storage, Wh |
| $Cb$ | Battery cost, USD |
| $Ct$ | Total cost, USD |
| $CP$ | Turbine power coefficent |
| $DCC$ | Direct capital cost, USD |
| $DOD$ | Depth of discharge |
| $Dr$ | Rotor diameter, m |
| $FF$ | Fouling factor |
| $FOBc$ | Full over board cost, USD |
| $Gb$ | Solar flux, $W/m^2$ |
| $HPP$ | High pressure pump |
| $Hh$ | Hub height, m |
| $HWT$ | Horizontal wind turbine |
| $k$ | Permeability |
| $M$ | Mass flow rate, $m^3/h$, kg/s |
| $n$ | Number, # |
| $ne$ | Element number |
| $NV$ | Number of pressure vessels |
| $NOB$ | Number of batteries, # |
| $NOC$ | Number of cells, # |
| $NOM$ | Number of modules, # |
| $NWT$ | Number of wind turbines, # |
| $OH$ | Operating hours, h |
| $P$ | Power, Permeator, or Pressure, bar |
| $Pm$ | Module power, W |
| $Pt$ | Total power, W |
| $Pw$ | Wind power, kW |
| $PV$ | Photovoltaic |
| $\Delta P$ | Pressure, bar |
| $RR$ | Recovery ratio |

| | |
|---|---|
| *RPMr* | Rotor speed, rpm |
| *SPC* | Specific power consumption, kWh/m$^3$ |
| *SR* | Salt rejection |
| *T* | Temperature, °C |
| *Tor* | Torque, Nm |
| *TCF* | Temperature correction factor |
| *UPC* | Unit product cost, USD/m$^3$ |
| *V* | Volt |
| *Vws* | Start wind speed, m/s |
| *Vwa* | Average wind speed, m/s |
| *X* | Salinity, ppm |
| ***Subscripts*** | |
| *air* | Ambient |
| *b* | Brine, battery |
| *c* | Cell |
| *d* | Distillate product |
| *e* | Element |
| *f* | Feed |
| *m* | Module |
| *ORC* | Organic rankine cycle |
| *RO* | Reverse osmosis |
| *t* | Turbine, total |
| *v* | Vessel |
| *w* | Water |
| ***Greek*** | |
| *η* | Efficiency, % |
| Π | Osmotic pressure, kPa |
| *ρ* | Density, kg/m$^r$ |
| *ω* | Rad/s |

## Appendix A

### A: The RO Model

The mathematical model for the proposed RO unit is written as follows [2,24]:

The feed flow rate $M_f$ based on recovery ratio *RR* and distillate flow rate $M_d$ is

$$M_f = \frac{M_d}{RR} \tag{A1}$$

The distillate product salt concentration $X_d$ is

$$X_d = X_f \times (1 - SR) \tag{A2}$$

where $X_f$ is the feed flow rate salt concentration, and *SR* is the salt rejection percentage. The rejected brine is found from

$$M_b = M_f - M_d \tag{A3}$$

The rejected salt concentration kg/m$^3$ is estimated by

$$X_b = \frac{M_f \times X_f - M_d \times X_d}{M_b} \tag{A4}$$

The average salt concentration kg/m$^3$ is estimated as

$$X_{av} = \frac{M_f \times X_f + M_b \times X_b}{M_f + M_b} \tag{A5}$$

The temperature correction factor *TCF* is found by

$$TCF = exp\left(2700 \times \left(\frac{1}{273 + t} - \frac{1}{298}\right)\right) \tag{A6}$$

Membrane water permeability $k_W$ is

$$k_w = 6.84 \times 10^{-8} \times (18.6865 - (0.177 \times X_b))/(t + 273)) \tag{A7}$$

The salt permeability $k_s$ is

$$k_s = FF \times TCF \times 4.72 \times 10^{-7} \times \left(0.06201 - \left(5.31 \times 10^{-5} \times (t + 273)\right)\right) \tag{A8}$$

where *FF* is the membrane fouling factor. The calculations of osmotic pressure for the feed, brine, and distillate product are found as

$$\Pi_f = 75.84 \times X_f \tag{A9}$$

$$\Pi_b = 75.84 \times X_b \tag{A10}$$

$$\Pi_d = 75.84 \times X_d \tag{A11}$$

The average osmotic pressure on the feed side is

$$\Pi_{av} = 0.5 \times \left(\Pi_f + \Pi_b\right) \tag{A12}$$

The net osmotic pressure across the membrane is

$$\Delta\Pi = \Pi_{av} - \Pi_d \tag{A13}$$

The net pressure difference across the membrane is

$$\Delta P = \left(\frac{M_d}{3600 \times TCF \times FF \times A_e \times n_e \times N_v \times k_w}\right) + \Delta\Pi \tag{A14}$$

where $A_e$ is the element area in m$^2$; $n_e$ is number of membrane elements; and $N_v$ is the number of pressure vessels. The required power input in kW for the RO high pressure pump (HPP) is estimated as

$$HPP_{power} = \frac{1000 \times M_f \times \Delta P}{3600 \times \rho_f \times \eta_p} \tag{A15}$$

where $\rho_f$ is the feed flow rate density, and $\eta_p$ is the driving pump mechanical efficiency. The specific power consumption in kWh/m$^3$ is estimated as

$$SPC = \frac{HPP_{power}}{M_d} \tag{A16}$$

**B: The HWT Model**

The Horizontal Wind Turbine (HWT) as a part of wind desalination library (REDS [25]) was modeled according to the specification data obtained from the manufacture manual for many watt points, which vary from 0.5 to 8000 kW according to many companies. The data were obtained from more than 50 companies involved in wind turbine manufacturing. A developed model by Sharaf [26–29] was presented and correlated as a function of wind turbine power (*P*) as follows:

The starting wind speed m/s as a function of turbine power (kW):

$$Vw_s = 13.37 \times \epsilon^{(1.698^{-5} \times P)} - 10.72 \times \epsilon^{(-0.003214 \times P)} \tag{A17}$$

The average wind speed m/s:

$$Vw_a = 9.378 \times \left(P^{0.09862}\right) \tag{A18}$$

The rotor diameter m:

$$Dr = 2.573 \times \left(P^{0.4414}\right) \tag{A19}$$

The tower (Hub) height m:

$$Hh = 1.437 \times \left(P^{0.5046}\right) + 5.354 \tag{A20}$$

Air density kg/m$^3$ is calculated based on air and pressure temperature:

$$\rho_{air} = \frac{P_{air} \times 100}{0.287 \times (T_{air} + 273.15)} \tag{A21}$$

where $P_{air}$ is in bar and $T_{air}$ is in °C.

The rotor swept area m$^2$ is then calculated based on the rotor diameter $Dr$:

$$Ar = \pi \times (Dr/2)^2 \tag{A22}$$

The air mass flow rate kg/s is then calculated based on the density, rotor swept area and average wind speed:

$$M_{air} = \rho_{air} \times Ar \times Vw_a \tag{A23}$$

The required wind power kW:

$$P_w = \frac{\left(\frac{1}{2} \times \rho_{air} \times Ar \times \left(Vw_a{}^3\right)\right)}{1000} \tag{A24}$$

The power coefficient is calculated from the assigned power $P$ and the aerodynamic power $Pw$:

$$CP = \frac{P}{P_w} \tag{A25}$$

The rotor speed in rpm:

$$rpm_r = 347.6 \times \left(P^{-0.2909}\right) - 16.91 \tag{A26}$$

The rotor torque *Tor* in Nm based on the power of the turbine and the angular velocity ($\omega$):

$$\omega = \frac{(2 \times \pi \times RPM_r)}{60} \tag{A27}$$

$$Tor = \frac{(1000 \times P)}{\omega} \tag{A28}$$

The number of wind turbines can be calculated related to the total demanded power (*TP* kW) from the wind farm:

$$NWT = \frac{TP}{P} \tag{A29}$$

### C: The PV Model

PV system was considered a very important power source in this work. It was modeled according to the actual data presented through more than 150 data points from the manufacturing manuals. The range of the operating modules type was from 5 to 280 W. Each module watt type can calculate the module specification based on the data fed in the table. Table A1 illustrates the inputs and outputs of the developed lookup table model

block. SDS program [16,26] library is used to model and visualize the PV system program. The developed code is introduced to calculate the following:

**Table A1.** PV input and calculated parameters by the use of SDS program [25].

| Inputs: | Outputs: |
|---|---|
| 1—Operating hours (*OH*), h<br>2—Solar flux ($G_b$), W/m$^2$<br>3—Number of cloudy days factor<br>4—System total power ($P_t$), kW<br>5—Module power ($P_m$) (5–280 W)<br>6—Battery depth of discharge (*DOD*)<br>7—Battery voltage ($V_b$), Volt<br>8—Battery efficiency, %<br>9—Battery unit price ($C_b$), USD | 1—Open circuit voltage ($V_{oc}$), Volt<br>2—Short circuit current ($I_{sc}$), A<br>3—Maximum voltage ($V_m$), Volt<br>4—Maximum current ($I_m$), A<br>5—Cell and Module efficiencies, %<br>6—Net weight, kg<br>7—The dimensions, m$^2$<br>8—Module price, USD/W<br>9—Number of cells and modules (*NOC*)<br>10—Cell area ($A_c$), cm$^2$<br>11—Module area ($A_m$), m$^2$<br>12—Total system area ($A_t$), m$^2$<br>13—Battery storage, Wh<br>14—Battery capacity, Ah<br>15—Number of batteries (*NOB*)<br>16—Full over board cost ($FOB_c$), USD. |

When calculating the main specifications (Table A1, parameters from 1 to 9) based on the module power, the following code was easily calculated. The number of modules (*NOM*) based on total power and module power:

$$NOM = \frac{P_t}{P_m} \tag{A30}$$

The module area in m$^2$ is then calculated based on module power $P_m$ and efficiency $\eta_m$:

$$A_m = 100 \times \frac{P_m}{G_b \times \eta_m} \tag{A31}$$

Then the total area in m$^2$ was calculated:

$$A_t = A_m \times NOM \tag{A32}$$

The cell area in cm$^2$ based on the number of cells (*NOC*) that been calculated from the lookup table.

$$A_c = \frac{A_m \times 10^3}{NOC} \tag{A33}$$

The battery storage in Wh based on the operating hours (*OH*), number of cells (*NOC*), the total power ($P_t$), battery efficiency and depth of discharge (*DOD*):

$$BS = \frac{OH \times NOC \times P_t}{DOD \times \eta_b} \tag{A34}$$

If a 24 V system were chosen, the required (*AH*) of batteries would be 16,585/24,700.

$$AH = \frac{BS}{V_m} \tag{A35}$$

Number of batteries can be calculated as follows based on the maximum voltage and the battery voltage:

$$NOB = \frac{V_m}{V_b} \tag{A36}$$

The system total costs in ($C_t$, *USD*) are then calculated based on the full over board costs of the modules ($FOB_c$) and the batteries costs ($C_b$):

$$C_t = (P_t \times FOB_c) + (C_b \times NOB) \tag{A37}$$

where the $FOB_c$ includes cables, connections, worker time, inverter unit, and maintenance costs.

**D: Monthly average irradiation and wind speed**

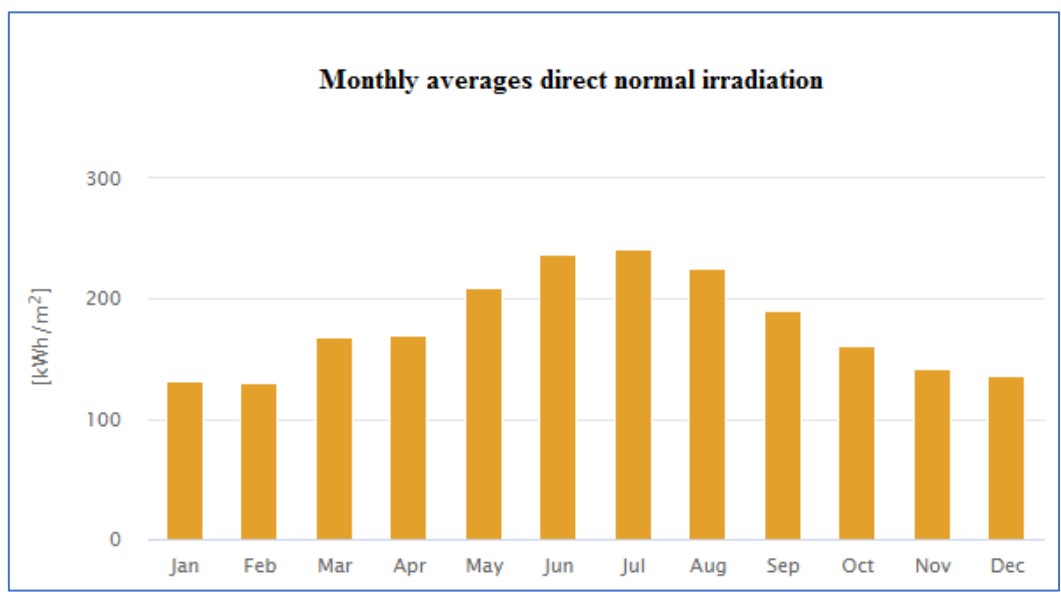

**Figure A1.** Monthly average irradiation for Suez Gulf region.

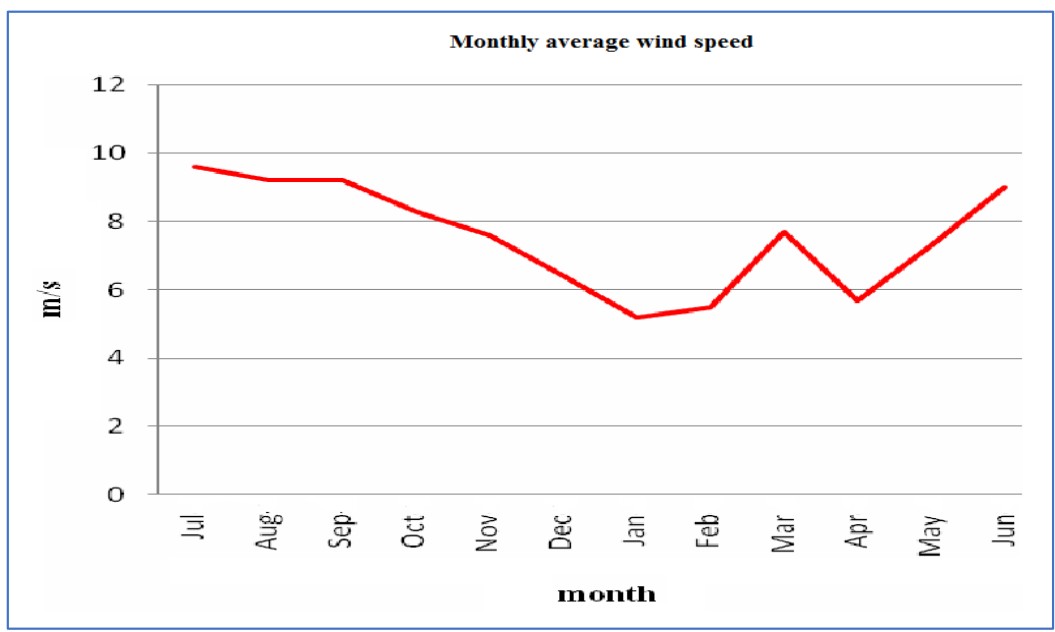

**Figure A2.** Monthly average wind velocity for Suez Gulf region.

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
