# Peer review of "Techno-Economic Optimization of a Solar–Wind Hybrid System to Power a Large-Scale Reverse Osmosis Desalination Plant"

_sustainability, doi:10.3390/su132011508_

Round 1
Reviewer 1 Report
The paper titled “Techno Economic Optimization of Solar-Wind Hybrid Systems Powered Large Scale Reverse Osmosis Desalination Plants” and written by A. M. Soliman et al. is reports a study base on simulation about optimization of reverse osmosis desalination plants powered by renewable energy. The paper needs a major revision based on the following comments:
- The English should be revised.
- Please, provide a space between de number and the units, for example 200 W instead of 200W
- This manuscript is about brackish water, seawater or both? If it is only for one of them, it should be considered in the title.
- Page 1, the sentence that ends in line 32. I would provide a citation the support the mentioned statement, I suggest the following one:
- Reverse osmosis desalination: A state-of-the-art review
- Page 1, line 33. Authors meant desalination instead of distillation right? And please, write mechanical without capita letter.
- Page 1, line 36. There are desalination plants with a larger capacity than 300,000 m3/d. Please, revise the literature.
- Page 1, line 41. Reverse osmosis was already abbreviated, so please, use the abbreviation RO. Check the entire document.
- Page 1, line 44, Why the authors wrote solar batteries, it should be just batteries, right? It is supposed the that batteries work with electricity and the PV with solar radiation, no the batteries directly with solar radiation.
- Page 2, line 49. Please, write h instead of hrs. Check the entire document.
- Page 2, line 55. There is extra space before Ahmed…Please, revise the entire document. Page 2, line 60. Use the abbreviation RO.
- In the section introduction. I miss few sentence about the operation of RO desalination plants powered by renewable energies. For example, Desalination plant can work under steady regime or variable regime as the energy source is variable. The authors should mention these mode of operations by commenting and citing some works about this issue. I recommend the following papers that deal with variable operation and even the loss of performance due to membrane fouling after working under intermittent operation:
- Performance evaluation and boron rejection in a SWRO system under variable operating conditions
- Reverse osmosis (RO) membrane desalination driven by wind and solar photovoltaic (PV) energy: State of the art and challenges for large-scale implementation
- On-Off Control Strategy in a BWRO System under Variable Power and Feedwater Concentration Conditions
- Theoretical performance prediction of a reverse osmosis desalination membrane element under variable operating conditions
- Long-term intermittent operation of a full-scale BWRO desalination plant
- Performance evaluation of a brackish water reverse osmosis pilot-plant desalination process under different operating conditions: Experimental study
- Page 2, line 70. Please, write 3 as superscript.
- Page 2, lines 93 and 94. I think the first sentence should be removed. It does not provide any information about the methods.
- Page 2, line 97. Please write Figure 1 instead of Figure (1) according with the style of the journal. And in singular, it does not matter if Figure 1 has subfigures.
- Page 6, line 158. Use only the abbreviation UPC.
- Please, write W instead of Watt. Revise the entire document.
- Could the authors provide the profiles of wind and solar radiation?
Author Response
|
Comments |
Response |
|
Done (by MDPI English Editing) |
|
Done |
|
For both |
|
Done |
|
Done |
|
3000,000 m3/d |
|
Done |
|
Done |
|
Done |
|
Done |
|
Done |
|
Done |
|
Done |
|
Done |
|
Done |
|
Done |
|
In appendix D |

Reviewer 2 Report
A very interesting article by Slimon et al. comparing solar photovoltaic (PV), horizontal wind turbine (HWT) and PV-HWT hybrid systems to run the reverse osmosis desalination process. The comparison was made based on technique and economics. Optimization methods were used to minimize cost. The study recommends PV-powered RO system. Overall, the flow of the manuscript is logical. The manuscript requires minor revision before considering this manuscript for publication:
- A brief of summary of various assumptions used in different analyses should be listed.
- In some figures, e.g., 7 and 8, the unit of area should be m2, not m2.
- Comparable literature studies should be cited and discussed in the manuscript.
- Include a section on recommendations before the conclusions section.
- English needs improvement. Consult a native speaker for corrections and improvements.
- The article needs proofreading.
Author Response
|
Comments |
Responses |
|
The summary have been listed in tables |
|
Done |
|
3. Comparable literature studies should be cited and discussed in the manuscript. |
Fig. 9 |
|
اsection 6 |
|
Done (by MDPI English Editing) |
|
Done (by MDPI English Editing) |

Round 2
Reviewer 1 Report
Authors have adressed all my comments
This manuscript is a resubmission of an earlier submission. The following is a list of the peer review reports and author responses from that submission.